# Llamas (mostly) think in English: On Causal Interventions in the Latent Language of Transformers

## Abstract

Previous research on the Llama-2 family of Large Language Models (LLMs) suggested a correlation indicating the use of English as a intermediary language within these models for tasks in non-English languages. We improve on this by demonstrating a causal relationship. By intervening on the intermediate layers during a forward pass, we show that projecting out the activations onto a subspace corresponding to the correct prediction in English impairs the model's ability to make correct predictions on non-English translation tasks. Projecting out an unrelated English subspace, or a related subspace in a non-English language, has little effect, demonstrating that this family of models store concepts that have a high similarity to the corresponding concept in English in the residual stream.

## 1 Introduction

The dramatic abilities of Large Language Models (LLMs) using the transformer architecture Vaswani et al. (2017); Phuong & Hutter (2022) are rather surprising, given the sole goal during training is to predict the next word in a sequence. LLMs generalize to many out-of-distribution tasks, and exhibit abilities typically associated with intelligence, such as solving difficult maths problems, tool use, and demonstrating theory of mind Bubeck et al. (2023). Interestingly, models predominantly trained on English data tend to perform well in other languages, even when other languages constitute a tiny proportion of the training data K et al. (2020); Blevins & Zettlemoyer (2022). Tianyi Tang et al. (2024) show that language-specific neurons in LLMs are responsible for their multilingual capabilities, and that activating or deactivating these neurons can control the output language. Additionally, Julen Etxaniz et al. (2023) show that LLMs trained predominately in English can perform better on tasks in non-English by explicitly prompting the model to translate to English, solving the task in English, and then translating back to the target language. Shi et al. (2022) show that by using chain-of-thought Chu et al. (2023) prompting, models can perform vastly better on even obscure languages.

### 1.1 Llama works in English Wendler et al. (2024)

Wendler et al. (2024) claim that the Llama-2 family of multilingual transformers "work in English" by showing that on translation tasks between non-English languages, the model assigns a high probability to the corresponding answer in English midway through a forward pass. More precisely, given the multi-shot translation prompt[1] used by Wendler et al. (2024) from French (*Français*) to Chinese (中文),

Français: " vertu" - 中文: "德"
Français: " siège" - 中文: "座"
Français: " neige" - 中文: "雪"
Français: " montagne" - 中文: "山"
Français: " fleur" - 中文: "

---

[1] We modify the prompt provided in Wendler et al. (2024) slightly and prepend spaces to words using the latin alphabet to aid with tokenization.

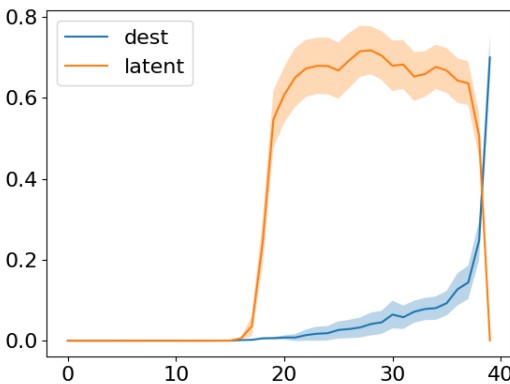

Figure 1: The probability assigned both to the correct answer in Chinese, and in the latent language English when translating from French to Chinese. Probability measured for every layer using *logit lens* for Llama-2-7b. We sum the probability mass over not just the exact correct answer in Chinese, but other semantically similar answers as well.

the model, using in-context-learning Brown et al. (2020) predicts the next token as 花, the correct translation of `fleur` (flower) from French to Chinese. Surprisingly, if the technique of *logit lens* Nostalgebraist (2020) is used, and the residual stream midway through a forward pass is passed through the unembedding stage of the model, a high probability is assigned to the corresponding token `flower` in English (Figure 1). We refer to the language being translated from as the *source* (here, French) the language being translated to as the *target* (Chinese) and the language we measure the probability mid-way through the forward pass as the *latent* language.

This effect is observed regardless of which non-English languages are chosen for the source and target, and only if the latent language is English. This is suggestive that the model is using English as an intermediary language for translation tasks. However, this is only a correlation, and may just be an artefact of tokenization, or that the vocabulary of the model is predominantly English.

## 1.2 MAIN CONTRIBUTION

For translation tasks between non-English languages on the Llama-2-7b model Touvron et al. (2023):

- We show a causal relationship between the unembedding vectors of the corresponding answer in English and the downstream prediction of the model in the target language. By computing the vector projection $\text{proj}_{\mathbb{S}}(h_i)$ of the intermediate activations $h_i$ onto a subspace $\mathbb{S}$ corresponding to the correct prediction in English, and subtracting this projection out, this impairs the model's ability to make correct predictions on non-English translation tasks. This effect is far weaker when projecting out a related subspace in a non-English language, and not present at all when projecting out an unrelated subspace (using words unrelated to that being translated), regardless of the language chosen to intervene with.

- Normally, steering vectors Turner et al. (2023) are obtained by computing a forward pass over inputs of interest, and extracting vectors of interest from the activations. We show a weak effect demonstrating that on translation tasks, we can cause the model to predict a desired counterfactual translation into the target language by steering directly using the unembedding vectors of the corresponding word in English. Note that this effect is not robust, and can in may cases cause the model activations to diverge during a forward pass.

## 2 PRELIMINARIES

### 2.1 MOTIVATION

LLMs are impressive in the capabilities that they have, but it is not at all clear how they can perform the tasks they do. After training, the mechanism by which the model can make good predictions is present, but buried amongst the weights of the network.

This motivates a more transparent approach to understanding the behaviour of models, and what the field of *mechanistic interpretability* aims to achieve: A holistic understanding of how models can perform particular tasks. Meng et al. (2022) show how to extract and modify particular facts known by the model. Wang et al. (2022) demonstrate how particular attention heads in a transformer model can learn how to solve simple in-context-learning tasks. Li et al. (2023) demonstrate that LLMs internally construct models of the world to solve tasks. However, works like this are very costly in terms of human time spent on analysis of models, and it is desirable to find more automatic approaches to understanding models. Zhang & Nanda (2024) introduce *activation patching*, to take the activations from a forward pass on corrupted input, patch them into the activations of a forward pass on clean input, and observe the causal downstream effect. Todd et al. (2024) present *function vectors*, a direction in latent space representing a particular function or operation, and use this to steer the model on other such inputs.

### 2.2 TRANSFORMERS

The transformer architecture Vaswani et al. (2017) is a deep neural network trained on a large corpus of text data, with the goal Radford et al. (2019); Brown et al. (2020) of predicting the next token in a sequence. The training data is composed of strings of text, converted into a sequence of *tokens* $t_1, \ldots, t_{\text{seq}}$ of length seq, where each token is a word or subword obtained using *byte pair encoding* Sennrich et al. (2016). Each token is an integer in the range $t_i \in \{1, \ldots, d_{\text{vocab}}\}$, where $d_{\text{vocab}}$ is the size of the *vocabulary* for the model.

The model then converts these tokens into embeddings $e^{t_1}, \ldots, e^{t_{\text{seq}}} \in \mathbb{R}^{d_{\text{model}}}$, by using tokens to index rows in the embedding matrix $\boldsymbol{E} \in \mathbb{R}^{d_{\text{vocab}} \times d_{\text{model}}}$. Stacking these embeddings gives the first hidden state $\boldsymbol{H}^1 \in \mathbb{R}^{\text{seq} \times d_{\text{model}}}$.

$$e^i = \boldsymbol{E}_{t_i,:} \qquad \boldsymbol{H}^1_{i,:} = e^{t_i} \tag{1}$$

These embeddings are then passed through a series $T_1, T_2, \ldots, T_n$ of $n$ layers called *transformer blocks*. For each hidden state $\boldsymbol{H}^k$, the subsequent hidden state $\boldsymbol{H}^{k+1}$ is computed as $\boldsymbol{H}^{k+1} = T_k(\boldsymbol{H}^k)$, giving us a sequence of hidden states, or activations, $\boldsymbol{H}^1, \ldots, \boldsymbol{H}^{n+1}$. Following Elhage et al. (2021), we refer to the sequence of hidden states as the *residual stream*.

Specific to the Llama-2 family of models Touvron et al. (2023), the transformers blocks $T_i$ are defined as[2]

$$T_k(\boldsymbol{H}^k) = \boldsymbol{Z}^k + \text{GLU}(\text{RMS}(\boldsymbol{Z}^k))$$
$$\text{where } \boldsymbol{Z}^k = \boldsymbol{H}^k + \text{MHA}(\text{RMS}(\boldsymbol{H}^k))$$

where GLU is a Gated Linear Unit Shazeer (2020) using the Swish activation function, MHA is the multi-head self-attention mechanism Vaswani et al. (2017), and RMS is the Root Mean Square normalization operation Zhang & Sennrich (2019). Each of the internal components of $T_k$ have a set of learned weights specific to that layer.

The final hidden state $\boldsymbol{H}^{n+1}$ is then RMS normalized, and then multiplied by the unembedding matrix $\boldsymbol{U} \in \mathbb{R}^{d_{\text{vocab}} \times d_{\text{model}}}$ to obtain the *logits* $\boldsymbol{L} \in \mathbb{R}^{\text{seq} \times d_{\text{vocab}}}$:

$$\boldsymbol{L} = \boldsymbol{U}(\text{RMS}(\boldsymbol{H}^{n+1})) \tag{2}$$

which are finally passed through a *softmax* Goodfellow et al. (2016) operation to obtain a set of probability distributions $\boldsymbol{P} \in \mathbb{R}^{\text{seq} \times d_{\text{vocab}}}$, representing (for a given prefix $t_1, \ldots, t_i$ of the input

---

[2]We omit the details of positional embeddings. Llama-2 uses Rotary Positional Embedding (RoPE) Su et al. (2023) which are performed inside the self-attention mechanism.

sequence) the model's prediction for the next token $t_{i+1}$.

$$\boldsymbol{P}_{i,j} = \frac{\exp(\boldsymbol{L}_{i,j})}{\sum_{j'} \exp(\boldsymbol{L}_{i,j'})} \approx \Pr(t_{i+1} = j | t_1, \ldots, t_i) \tag{3}$$

During inference we are only interested in the probability of the next unknown token $t_{\text{seq}+1}$, given by the last row $\boldsymbol{p} := \boldsymbol{P}_{\text{seq},:} \in \mathbb{R}^{d_{\text{vocab}}}$, derived from the last logit vector $\boldsymbol{l} := \boldsymbol{L}_{\text{seq},:} \in \mathbb{R}^{d_{\text{vocab}}}$.

We do not concern ourselves with the internal details of the self-attention mechanism, but focus only on interventions that modify the hidden states $\boldsymbol{H}^1, \ldots, \boldsymbol{H}^{n+1}$ between transformer blocks.

## 2.3 LOGIT LENS

The same argument can be made for transformers, which are also deep models with skip connections, so it is reasonable to expect that the intermediary activations have an interpretable relationship to the final logits (see Appendix A.1). Based on this idea, Nostalgebraist (2020) introduced *logit lens*: During a forward pass, the hidden states $\boldsymbol{h}^1 := \boldsymbol{H}^1_{\text{seq},:}, \ldots, \boldsymbol{h}^{n+1} := \boldsymbol{H}^{n+1}_{\text{seq},:}$ associated with the last token position/prediction of the next token are cached, and then fed through the final unembedding stage of the model (comprised of an RMS normalization layer, followed by multiplying by the unembedding matrix $\boldsymbol{U}$) to get increasingly better estimates[3] $\hat{\boldsymbol{l}}^1, \ldots, \hat{\boldsymbol{l}}^{n+1}$ of the logits $\boldsymbol{l}$, from which we can recover estimates of $p$ using the softmax operation.

$$\boldsymbol{l} = \boldsymbol{U}\text{RMS}(\boldsymbol{h}_{n+1}) \approx \hat{\boldsymbol{l}}^k = \boldsymbol{U}\text{RMS}(\boldsymbol{h}^k) \tag{4}$$

$$\boldsymbol{p}_i = \frac{\exp(\boldsymbol{l}_i)}{\sum_{i'} \exp(\boldsymbol{l}_i)} \approx \frac{\exp(\hat{\boldsymbol{l}}^k[i,j])}{\sum_{j'} \exp(\hat{\boldsymbol{l}}^k[i,j'])} \tag{5}$$

# 3 EXPERIMENTS

## 3.1 SUBSPACE REJECTION

To demonstrate Llama-2's reliance on English as an intermediary language, we perform a series of interventions on the residual stream of the model during a forward pass on translation tasks. Information for prediction of the next token must be stored somewhere in the residual stream (it being the only causal path from earlier layers to later layers). We hypothesise that for translation, the information of the concept to translate is stored in a low-dimensional subspace $\mathbb{S} \subseteq \mathbb{R}^{d_{\text{model}}}$ of the residual stream, and this subspace is similar to a set $S = \{\boldsymbol{U}_{i_1,:}, \ldots \boldsymbol{U}_{i_m,:}\}$ of unembedding vector(s) $\boldsymbol{U}_{\cdot,:} \in \mathbb{R}^{d_{\text{model}}}$ for the token(s) describing that concept in English. If so, replacement of $\boldsymbol{h}^i$ with the orthogonal projection $\text{proj}_{\mathbb{S}^\perp}(\boldsymbol{h}^i)$ of $\boldsymbol{h}^i$ onto the orthogonal complement $\mathbb{S}^\perp$ of $\mathbb{S}$ should affect the model's ability to predict the correct answer in the target language.

$$\text{proj}_{S^\perp}(\boldsymbol{h}^i) = \boldsymbol{h}^i - \text{proj}_S(\boldsymbol{h}^i) = \boldsymbol{h}^i - S(S^T S)^{-1} S^T \boldsymbol{h}^i \tag{6}$$

As an abuse of notation, we write $\mathbb{S}$ as simply $S$, and define the subspace $S$ as $\text{span}(S)$. We call $\boldsymbol{h}^i_{\perp_S} := \text{proj}_{S^\perp}(\boldsymbol{h}^i)$ the *rejection* of $\boldsymbol{h}^i$ from $S$. We intervene on the hidden layers $\boldsymbol{h}^i$ between transformer blocks during a forward pass and replace each $\boldsymbol{h}^i \leftarrow \boldsymbol{h}^i_{\perp_S}$ during a forward pass over layers of interest. We then observe the downstream effect on the model's prediction on the target language. We construct two kinds of subspaces to reject, $S_{lang}$ and $S'_{lang}$ as follows:

- $S_{\text{lang}}(t_{\text{source}})$: We translate the correct prediction token $t_{\text{source}}$ from the source language to the corresponding token $t_{\text{lang}}$ in language *lang*. We consider many such valid translations for a given source word. For example, when translating *livre* (book) from French to English, we consider any of *book*, *manuscript*, *volume* or *tome* as valid translations. We also considered valid translations with and without leading spaces. For clarity, we write spaces as "␣". If the token $t_{\text{target}}$ is not present in the vocabulary (in that the tokenizer splits the token into subwords), we use only the first such token (under the assumption that if the model correctly predicts the first subtoken, it will correctly predict the rest Pal et al. (2023)).

---

[3]Note by definition that $\hat{\boldsymbol{l}}^{n+1} = \boldsymbol{l}$ is exact.

The subspace is then constructed as

$$S_{lang} = \{\boldsymbol{U}[t_{lang}] : t \simeq t_{lang}\} \cup \{\boldsymbol{U}[\_t_{lang}] : t \simeq t_{lang}\}$$

where $t \simeq t_{lang}$ denotes that $t$ and $t_{lang}$ are words with semantically identical or similar meanings, but in different languages. For example, given the source word for translation was *livre* (book), and potential translations of *livre* as *book*, *manuscript*, *volume* or *tome*, the correpsonding subspace would be[4]

$$S_{\text{en}}^{\text{livre}} = \text{Span}\{\boldsymbol{U}["book"], \boldsymbol{U}["\_book"], \boldsymbol{U}["volume"], \boldsymbol{U}["\_volume"],$$
$$\boldsymbol{U}["man"], \boldsymbol{U}["\_manuscript"], \boldsymbol{U}["t"], \boldsymbol{U}["\_t"]\}$$

Here, $\boldsymbol{U}[x]$ denotes the unembedding vector $\boldsymbol{U}[i_x]$, where $i_x$ is the index of the token $x$ in the vocabulary.

- $\bar{S}_{lang}(t_{\text{source}})$: Same as above, but we choose an incorrect translation $t_{\text{lang}} \not\simeq t_{\text{source}}$ in *lang* of the word $t_{\text{source}}$, together with other words with similar meanings to $t_{\text{lang}}$, also in language *lang*.

  Following the above example, we might choose the incorrect English translations *dog*, *canine* and *puppy* for *livre*, giving the corresponding subspace

  $$\bar{S}_{\text{en}}^{\text{livre}} = \{\boldsymbol{U}["dog"], \boldsymbol{U}["\_dog"], \boldsymbol{U}["can"], \boldsymbol{U}["\_can"], \boldsymbol{U}["pu"], \boldsymbol{U}["\_pu"]\}$$

  The incorrect translations were chosen by taking a dearrangement of the dataset, ensuring that the correct answer was not chosen, and that the word was not present in the translation prompt.

The subspaces are constructed in this way (with and without spaces) to deal with an artefact of the tokenization process: Often a word may appear in the vocabulary with or without a leading space (or sometimes both are present) and correspondingly, the model may choose to predict the word with or without a leading space.

Wendler et al. (2024) observed the phenomena where the model has three distinct phases during a forward pass: an initial phase where the model is not "thinking" in English, nor the target language (layers 1 to 19), a middle phase where the model is "thinking" in English (layers 20 to 29), and a final phase where the model is "thinking" in the target language (layers 30 to 32), as measured using the logit lens probability (Figure 1).

We perform the intervention over all layers inside the interval $[a, b]$, for some $1 \leq a < b \leq 32$. A sweep was performed over all possible intervals to perform the rejection intervention on with English as the latent language (see Figure 5a), but we found that the choice of layer was not critical, so long as it was sufficiently deep into the network to cover the region where the model "thinks in English".

The critical point for where the intervention starts working is around the transition from the initial to middle phase. Note that the rejection intervention does not completely destroy the model's ability to predict the correct answer, indicating that while the English subspace is important, it does not fully contain all the information required for prediction. So, we fix the rejection intervention to be performed on all layers, and compare to a baseline where no intervention is performed (Figure 2).

An alternative hypothesis is that the residual stream is fragile, and that any similar rejection intervention would have a similar effect. To test this, we perform the same intervention for all choices of source, latent and target languages (where the three languages are distinct), as well as performing the rejection for the unrelated subspace $S'_{\text{lang}}$.

We find that the rejection intervention has broadly no effect for latent languages that are not English (Figure 3), and no effect for unrelated subspaces in any language (Figure 4), which supports the hypothesis that we are not just damaging the residual stream in general, but performing a targeted intervention on the English subspace.

---

[4]Note that *tome* is not present in the vocabulary for Llama-2, so we only consider the first subtoken *t*, and similar for *manuscript*.

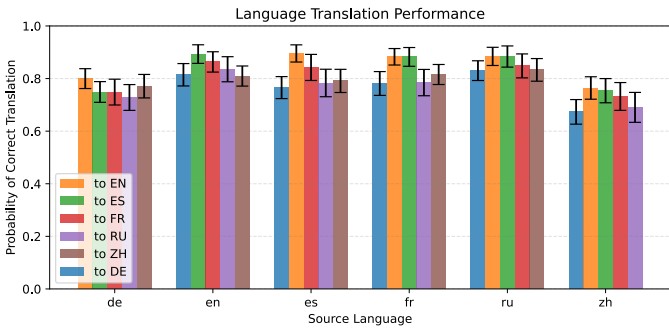

Figure 2: A plot of the average probability assigned to the correct answer in the target language when translating from the source language, for all pairwise translations between French (FR), German (DE), Chinese (ZH), English (EN), Spanish (EN) and Russian (RU). We see similar performance for all languages, with slightly worse performance when translating from Chinese. Error bars are 95% confidence intervals.

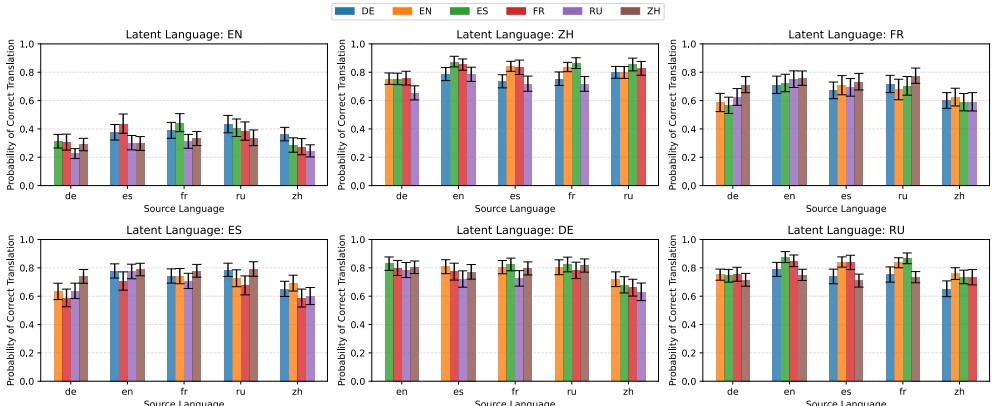

Figure 3: A plot of the average probability assigned to the correct answer in the target language when translating from the source language, and rejecting the latent language, for all possible triplets of (source, latent, target) languages. We see a stark drop in performance when rejecting the English subspace, but little effect for other languages. Error bars are 95% confidence intervals.

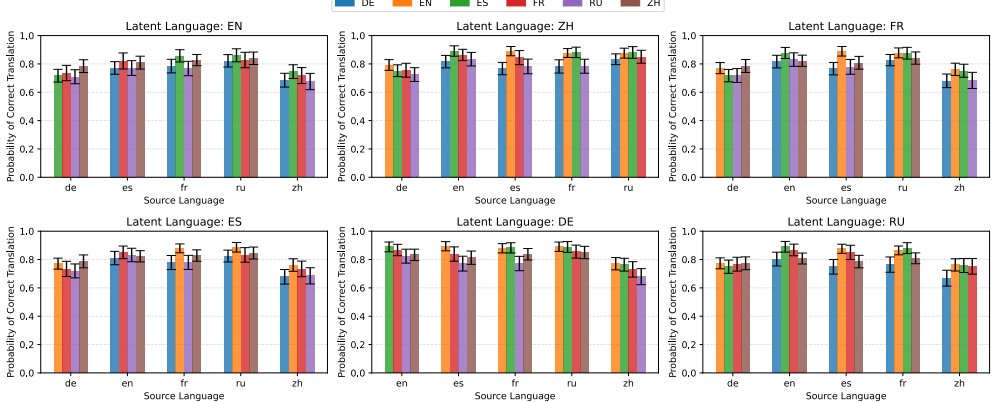

Figure 4: Same figure as Figure 3, but for the unrelated subspace. We see little effect on the model's ability to predict the correct answer. Error bars are 95% confidence intervals.

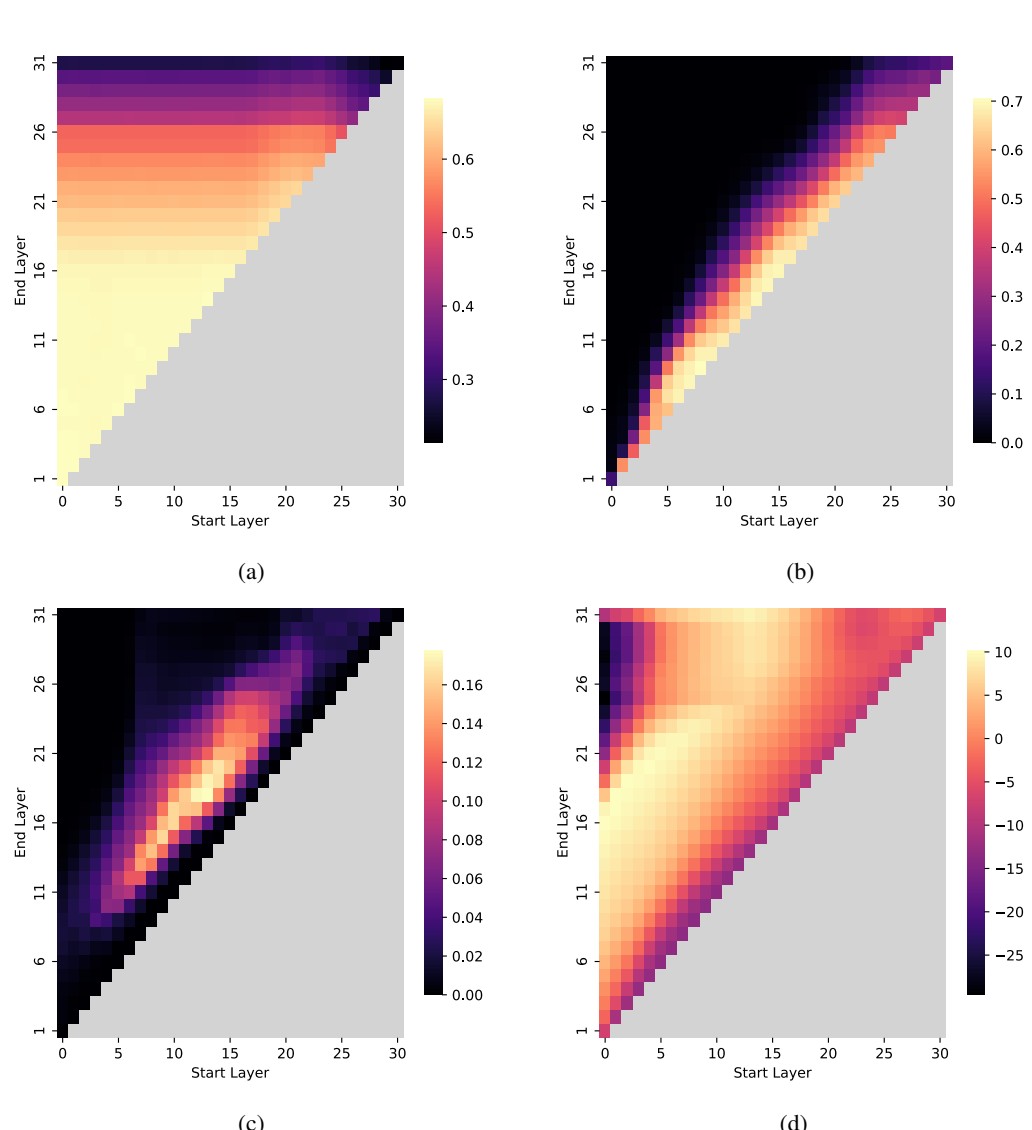

Figure 5: Translation tasks from source=French to target=Chinese, interventions in English.

(a) Probability assigned to correct answer with rejection intervention on related subspace (Section 3.1).

(b) Probability of correct answer with steering intervention from related subspace, to unrelated subspace (Section 3.2).

(c) Same as (b), but probability of desired *counterfactual* answer corresponding to the unrelated subspace we are steering towards (Section 3.2).

(d) *Log-probability* of the counterfactual answer, minus the log probability of the correct answer, using steering intervention. Positive values indicate the model is more likely to predict the counterfactual token over token corresponding to the correct translation.

Interventions performed between layers $[a, b]$, sweeping over all options for $0 \leq a < b < 32$ (zero-indexing layers), for model Llama-2-7b. Steering interventions use $c = 8$.

## 3.2 STEERING TOWARDS A COUNTER-FACTUAL TRANSLATION

The next experiment we perform is to try and steer the model to give a desired counterfactual translation in the target language only by modifying the activations using the unembedding vectors in the latent language. Given a source word $t_{\text{source}}$ to translate, and subspaces $S_{lang}(t_{\text{source}})$ and $\bar{S}_{lang}(t_{\text{source}})$ as described in Section 3.1, we perform the intervention

$$\boldsymbol{h}^i \leftarrow \boldsymbol{h}^i - \text{proj}_S(\boldsymbol{h}_i) + c\frac{1}{|\bar{S}|}\sum_{s\in\bar{S}} s \tag{7}$$

where $c > 0$ is a constant controlling the strength of the steering vector. We then observe the effect on the model's prediction by sweeping over layers to intervene on, as well as the size of the constant $c$.

We found that the strength of the effect grows monotonically with $c$, up to the point where the residual stream would diverge for $c \approx 10$. As expected, the effect was strongest when using $U_{\text{en}}$, but the choice of layer to intervene on is critical (and depends on the choice of $c$). We found the strongest effect (the one that boosts the probability of the counterfactual word in the target language the most) was to intervene on layers 13 to 18, and to choose $c = 8$. This is quite a large constant, as the norm of the vector that we project out is $\approx 4-5$, so essentially we are adding a larger vector back in to the one we projected out. For more reasonable values of $c$ (around 1-2), the effect was barely noticeable. This indicates that our steering intervention perhaps is not a suitable one, and that other methods should be investigated. We plot both the probability of the correct answer in the target language, and the probability of the counterfactual answer in the target language to demonstrate the effect is both a suppression of the correct answer, and a boosting of the counterfactual answer (Figure 5b).We also plot the difference in log-probabilities between the counterfactual and correct answer (Figure 5d). See Appendix A.1 for plots of sweeps over choices of $c$, and for various choices of languages.

## 3.3 CONCLUSION

Our results are somewhat mixed: We do confirm that to a degree, the LLama-2-7b model is using English as an intermediary language, and that the unembedding vectors in English do explain some of the model's behaviour in translation tasks. Projecting out in English clearly has a much stronger effect than projecting out in other languages, and the lack of effect for unrelated subspaces does indicate this intervention is not just causing general damage to the model.

However, the effect is not as strong as we would have hoped. There is still a lot of probability mass that the model assigns to the correct prediction after the rejection intervention. We would have liked to have seen the model's ability to predict the correct answer drop to near zero, but this was not the case. This indicates that the model is storing concepts elsewhere in a subspace that is not easily projected out, or that the model is able to recover from the intervention by using other information stored in the residual stream.

## 3.4 FUTURE WORK

We also performed the same experiment for Llama-2-13b, Gemma-2-2b and Gemma-2-9b, see Appendix A. While the same effect was observed for Llama-2-13b as was for Llama-2-7b, which was expected, the effect was much more destructive for the Gemma models. Any intervention, even those in an unrelated subspace or a non-English language, quite badly affected the model's ability to predict the correct answer. However, the Gemma models were still more vulnerable to rejections in English than any other language, causing the probability of prediction of the correct token to drop to near zero. We are unsure what to make of this behaviour, perhaps Gemma tends to store concepts in a less English-biased manner, or that the unembedding vectors are similar between languages, or that the Gemma models are more fragile in general. We leave this to future work.

The steering effect that we used was also rather brute-force, and would quite drastically change the activations midway through the network to something well outside the normal distribution. We would like to explore other methods of steering the model using the unembedding vectors only that are less destructive, while still being effective.

IMPACT STATEMENT

This paper presents work whose goal is to better understand the internals of Large Language Models, and how we can perform interventions on models in an interpretable fashion. We do not anticipate any ethical concerns arising from this work, as we are exploring the already existing capabilities of a pre-existing model.

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

# A  APPENDIX

### REPLICATION DETAILS

All code and datasets used for this paper can be found at [REDACTED]. Scripts are included to replicate the main results, as well as to generate the plots.

### DATASET DETAILS

In Wendler et al. (2024), the dataset was constructed by taking a corpus of words in English, and translating them externally into other languages using DeepL. A similarly constructed dataset was also used in the subsequent work by (Dumas et al.), which includes many more languages, as well as a list of other possible translations. For each non-English language *lang*, we took the English column and the *lang* column from their dataset, and prompted Claude 3.5 Sonnet as follows to create a high-quality set of additional possible translation from English to language *lang*. Here, `<x>` represents the translation of the English word x into language *lang*.

```
 en <lang>
book <book>
cloud <cloud>
...
power <power>
summer <summer>

Generate for me a CSV file, first column is the word in english,
second column is the <lang> translation provided, and third is a
list of synonyms of this word-pair in <lang>.  The list should be
formatted as a list of strings would be in python.  Do not force
the synonyms, they should all be a suitable replacement for the
word, or a word strongly associated.  For example, big and large
would be suitable synonyms.  Apple and pear would not.  Small and
tiny would be suitable.  Bag and handbag would be okay, as well
as fire and flame.  third, three and triple are all suitable.
Machine and car would not be.  Generate as many as you can for
each entry, but no more than 10.  If you can't find any, that's
okay, don't force words just to make the list long.  Some lists
can be short or even empty if the word is very unique and no
reasonable synonyms exist.
```

The original work included Japanese. We excluded as often the symbol for a word in Chinese and Japanese were identical (for example, *water* is expressed as 水 in both Chinese and Japanese), which would make translation artificially easy.

This gave us a dataset of words from English to Chinese, French, German, Russian and Spanish, and vice-versa. We try all 120 permutations of distinct source, latent and target languages, and perform the rejection intervention, both for the related subspace and the unrelated subspace.

We also slightly modify the prompt used by Wendler et al. (2024) (see Section 1.1) to deal with tokenization: we add a space character before the word if the language is non-Chinese (as the vocabulary of Llama-2-7b has many more tokens corresponding to a space character (e.g. `"_hello"`), followed by a word, than just the word itself), but we do not add the space for Chinese (Chinese tokens in the vocabulary are just the symbol itself, `"好"`, or the sequence of bytes in the UTF-8 representation of the symbol, `"<0xE5><0xA5><0xBD>"`). We ignored Chinese characters that could only be represented in the vocabulary as the raw UTF-8 sequence.

The prompt for each translation is fixed once and then used throughout the experiment. We use 4 words for the translation examples in the prompt, leaving the rest for translation. This was required as to allow for both kv-caching of the shared prefix, as well as to perform inference on a large number of words in parallel, to ensure the experiment was computationally feasible in a reasonable amount of time.

## A.1 Skip Connections

If models can make good predictions over a large class of domains, we would expect that somewhere in the residual stream of the transformer, the model stores useful information for prediction of the next token. In a standard feed-forward neural network, each pair of adjacent weight matrices have permutation invariances Entezari et al. (2022), so we should not expect the intermediary activations to have any clear resemblance to the final logits on the output.

It has been shown that deep vision neural networks pose a problem to train, due to gradients vanishing or exploding Pascanu et al. (2013). This was solved by adding skip connections He et al. (2016), which allowed very deep models to be trained effectively. As a side effect, transformers have been hypothesised to encourage the model to perform iterative inference: the intermediary activations represent the models' best current guess at the output midway through the forward pass, which gradually move towards the model's final prediction Jastrzębski et al. (2018). There are two factors at play here:

- The presence of skip connections in the transformer breaks permutation invariances, so the choice of basis for the intermediary activations is no longer arbitrary.
- Given some neural network $f_\Theta$ with a skip connection, the operation performed is $x + f_\Theta(x)$. The addition of L2 weight decay Goodfellow et al. (2016) during training encourages the weights of $f$ to be small, so all else equal, the model will prefer weights $\Theta$ such that $x + f_\Theta(x)$ is close to $x$.

## Reverse Lens

This technique had no improvement over adding the unembedding vectors directly into the residual stream (Section 3.2), but we include it (and the code) for posterity. *Tuned Lens* (TL) is a technique introduced by Belrose et al. (2023), which is identical to Logit Lens (LL), except for an extra learned linear mapping is applied to the hidden layers $h_i$ prior to normalization and unembedding. This gives a layer specific transformation $TL_i$ of the hidden state $h_i$. We can write the operation of Tuned Lens (and Logit Lens to compare) as

$$LL(\boldsymbol{h}^i) = \boldsymbol{U} RMS(\boldsymbol{h}^i)$$

$$TL_i(\boldsymbol{h}^i) = \boldsymbol{U} RMS(\boldsymbol{h}^i + A_i \boldsymbol{h}^i + \boldsymbol{b}^i)$$

$$RMS(x) = \frac{x}{\mu_{\boldsymbol{x}}} \odot \gamma$$

$$\mu_{\boldsymbol{x}} = \sqrt{\sum_{i=1}^{d_{\text{model}}} x_i^2 + \epsilon}$$

where $\boldsymbol{A}^i \in \mathbb{R}^{d_{\text{model}} \times d_{\text{model}}}$ and $\boldsymbol{b}^i \in \mathbb{R}^{d_{\text{model}}}$ are the learned parameters of the tuned lens for layer $i$, $\boldsymbol{U} \in \mathbb{R}^{d_{\text{vocab}} \times d_{\text{model}}}$ is the unembedding matrix, $\mu_{\boldsymbol{x}} \in \mathbb{R}$ is the scale factor for RMS norm, and

$\gamma \in \mathbb{R}^{d_{\text{model}}}$ is a learned parameter for RMS norm, taken from the final normalization layer of the transformer. Here, $\odot$ denotes elementwise multiplication. We do not concern ourselves with how the weights $\boldsymbol{A}^i, \boldsymbol{b}^i$ are learned, but make use of the pretrained weights for Tuned Lens available at AlignmentResearch (2023).

The idea behind tuned lens is that rather than unembedding $\boldsymbol{h}^i$ directly, an approximation $\boldsymbol{x} \mapsto \boldsymbol{x} + \boldsymbol{A}^i x + \boldsymbol{b}^i$ of the composition of the subsequent layers $\boldsymbol{x} \mapsto (T_n \circ T_{n-1} \ldots \circ T_i)(\boldsymbol{x})$ is learned and applied prior to normalization and unembedding, with the goal of aligning the semantics of the hidden state $\boldsymbol{h}^i$ with the input that $\boldsymbol{U}$ expects to operate on.

With this in mind, *Reverse Lens* is trying to run Tuned Lens backwards: Given a row of the unembedding matrix $\boldsymbol{U}_{i,:}$, what are the activations that Tuned Lens would map to this row? If Tuned Lens can recover the logits from the residual stream, then by running it backwards, we should be able to compute an approximation of the model's internal representation of $\boldsymbol{U}_{i,:}$ at any particular layer.

Unfortunately, RMS norm is not invertible, and $\boldsymbol{U}$ is not square, so we cannot directly invert Tuned Lens. To solve the first problem, note that for a fixed scale factor $\mu_{\boldsymbol{x}}$, RMS is a linear function, so we can factor $\mu_{\boldsymbol{x}}$ out, and the end result is the output logits will be scaled by $\mu_{\boldsymbol{x}}$. Since the argmax is invarant to scaling, we end up absorbing $\mu_{\boldsymbol{x}}$ elsewhere, and so we just set $\mu_{\boldsymbol{x}} = 1$ in the reverse operation. For the second, we do not require the ability to invert an arbitrary logit vector, but only vectors that correspond to a particular token that we wish to find the model's internal representation of. The logit $\boldsymbol{L}_{i,:}$ in the $i^{\text{th}}$ sequence position in the output is computed as $\boldsymbol{L}_{i,:} = (\boldsymbol{U}\text{RMS}(\boldsymbol{H}))_i \propto (\boldsymbol{U}_{i,:}) \cdot (\boldsymbol{H}_{i,:} \odot \gamma)$, the dot product of the $i^{\text{th}}$ row of $\boldsymbol{U}$ with the activations $\boldsymbol{H}$ (weighed by the RMS norm parameter $\gamma$). Since the dot product of a vector with itself is large, and the unembedding matrix $\boldsymbol{U}$ needs to be able to discriminate between many different tokens, we would expect the rows of $\boldsymbol{U}$ are approximately orthogonal. So, the vector that $\boldsymbol{U}$ would map to the standard basis vector[5] $\boldsymbol{e}_i$ is approximately $\boldsymbol{U}{:,i}$, the $i^{\text{th}}$ column of $\boldsymbol{U}$. We can then use this as the target for the reverse tuned lens operation.

We define the *reverse tuned lens* (RTL) for layer $i$ as follows: Taking as input an index $j \in \{1, \ldots, d_{\text{vocab}}\}$ and a cached scale factor $\mu \in \mathbb{R}$, we define the operation

$$\text{RTL}_i(j, \mu) = (\boldsymbol{I} - \boldsymbol{A}_i)^{-1}(\text{RMS}^{-1}(\boldsymbol{U}_{:,j}, \mu) - \boldsymbol{b}_i)\text{RMS}^{-1}(x, \mu) = \mu(x \cdot \gamma^{-1})$$

One can verify that $\text{argmax}_k(\text{TL}_i(\text{RTL}_i(j, \mu)))_k = j$ for all layers $i$, scaling factors $\mu$ and indices $j$, assuming that for every $i$, $\boldsymbol{U}_{i,:} \cdot \boldsymbol{U}_{j,:}$ is maximised when $j = i$.

However, when this was used in practice (repeating the experiment in Section 3.2 with RTL instead of the unembedding vectors directly), no appreciable difference in behaviour was observed. The rejection effect was equally effective as before. We include the code for the Reverse Tuned Lens operation, in the hope that perhaps it may be useful elsewhere.

FULL EXPERIMENTAL RESULTS FOR REJECTION

---

[5] Defined as $\boldsymbol{e}_i^i = 1$ and $\boldsymbol{e}_j^i = 0$ for $j \neq i$.

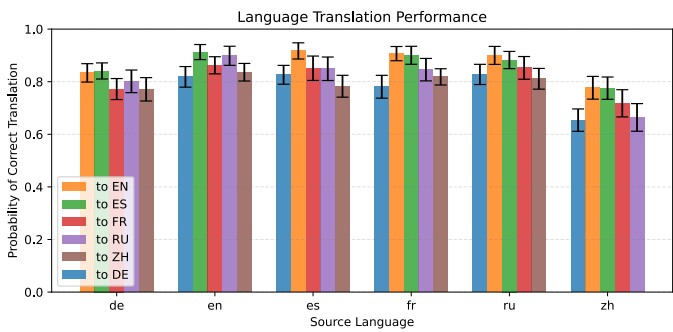

Figure 6: A plot of the average probability assigned to the correct answer in the target language when translating from the source language, for all pairwise translations between French (FR), German (DE), Chinese (ZH), English (EN), Spanish (EN) and Russian (RU). We see similar performance for all languages, with slightly worse performance when translating from Chinese. Error bars are 95% confidence intervals, model is Llama-2-13b.

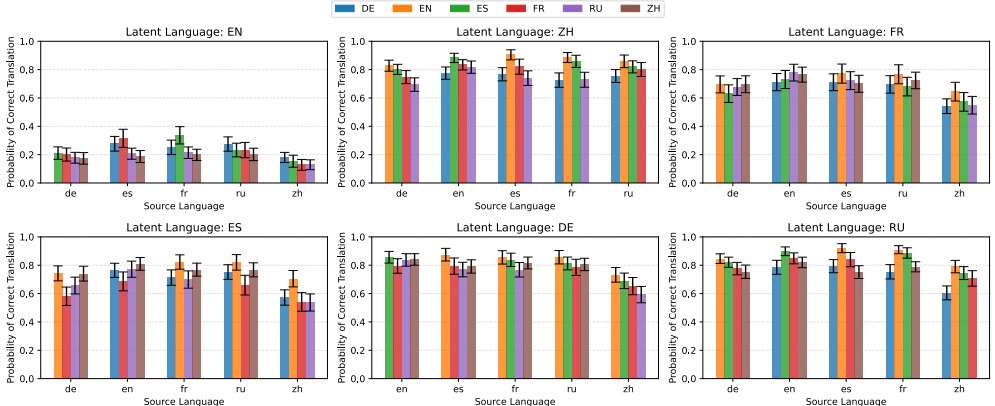

Figure 7: A plot of the average probability assigned to the correct answer in the target language when translating from the source language, and rejecting the latent language, for all possible triplets of (source, latent, target) languages. We see a stark drop in performance when rejecting the English subspace, but little effect for other languages. Error bars are 95% confidence intervals, model is Llama-2-13b.

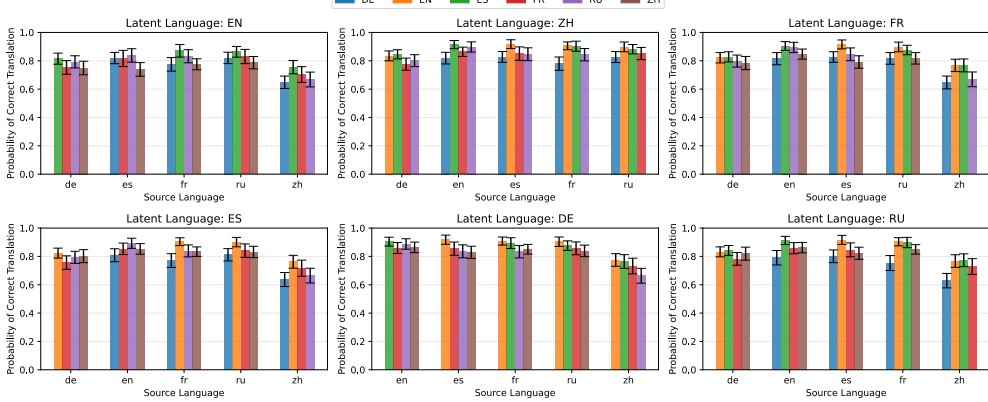

Figure 8: Same figure as Figure 3, but for the unrelated subspace. We see little effect on the model's ability to predict the correct answer. Error bars are 95% confidence intervals, model is Llama-2-13b.

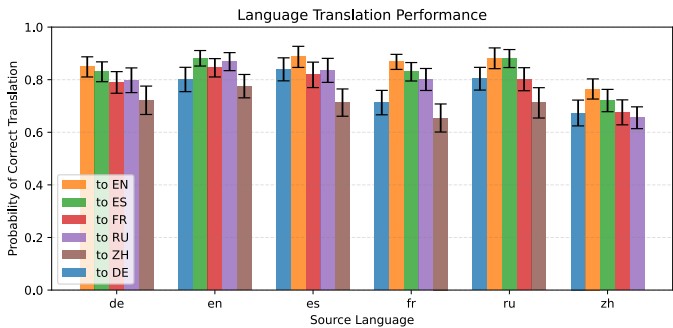

Figure 9: A plot of the average probability assigned to the correct answer in the target language when translating from the source language, for all pairwise translations between French (FR), German (DE), Chinese (ZH), English (EN), Spanish (EN) and Russian (RU). We see similar performance for all languages, with slightly worse performance when translating from Chinese. Error bars are 95% confidence intervals, model is Gemma-2-2b.

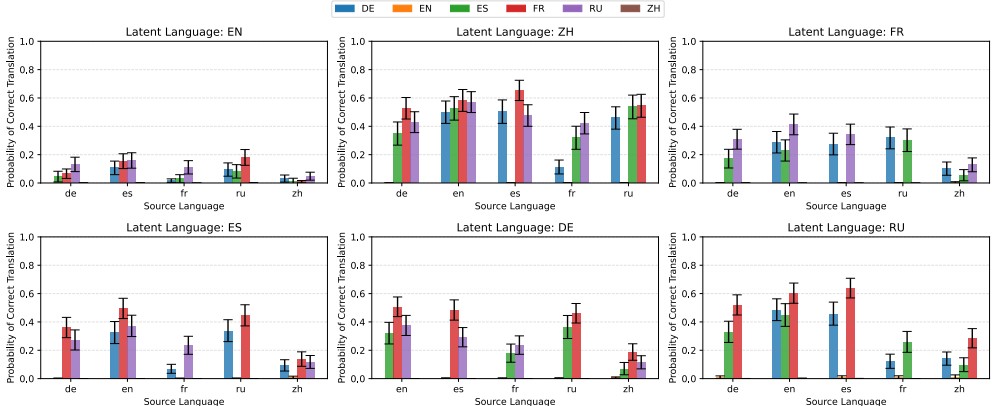

Figure 10: A plot of the average probability assigned to the correct answer in the target language when translating from the source language, and rejecting the latent language, for all possible triplets of (source, latent, target) languages. We see a stark drop in performance when rejecting the English subspace, but little effect for other languages. Error bars are 95% confidence intervals, model is Gemma-2-2b.

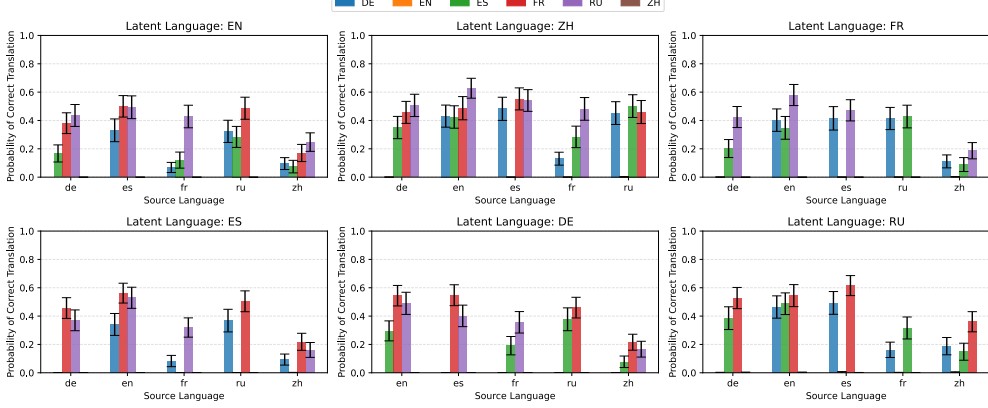

Figure 11: Same figure as Figure 3, but for the unrelated subspace. We see little effect on the model's ability to predict the correct answer. Error bars are 95% confidence intervals, model is Gemma-2-2b.

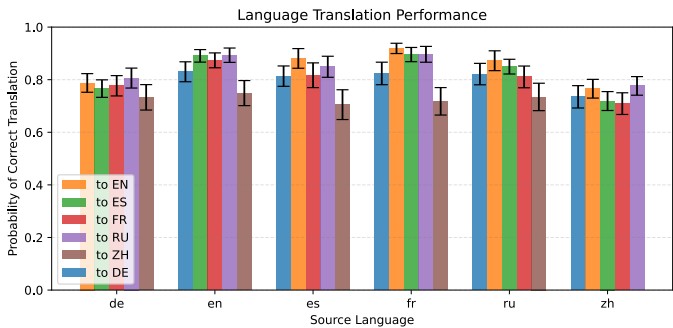

Figure 12: A plot of the average probability assigned to the correct answer in the target language when translating from the source language, for all pairwise translations between French (FR), German (DE), Chinese (ZH), English (EN), Spanish (EN) and Russian (RU). We see similar performance for all languages, with slightly worse performance when translating from Chinese. Error bars are 95% confidence intervals, model is Gemma-2-9b.

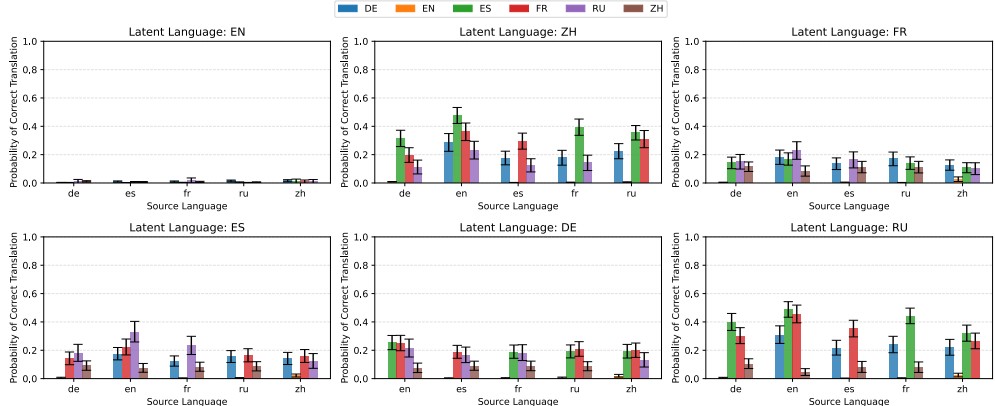

Figure 13: A plot of the average probability assigned to the correct answer in the target language when translating from the source language, and rejecting the latent language, for all possible triplets of (source, latent, target) languages. We see a stark drop in performance when rejecting the English subspace, but little effect for other languages. Error bars are 95% confidence intervals, model is Gemma-2-9b.

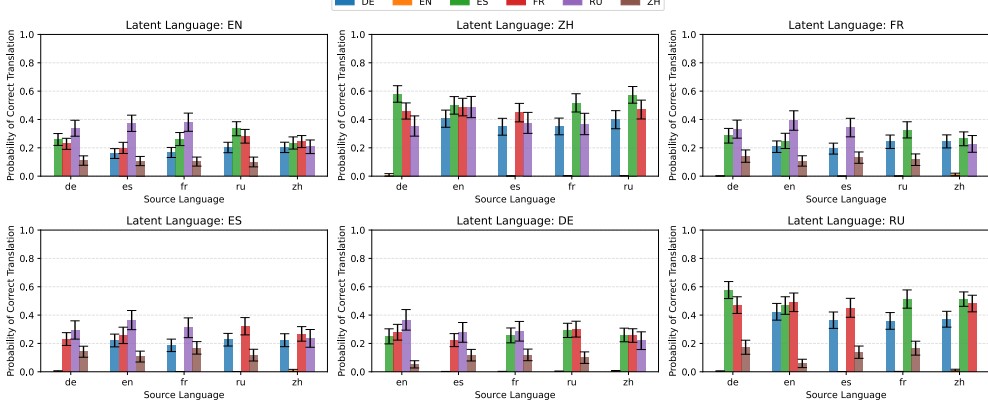

Figure 14: Same figure as Figure 3, but for the unrelated subspace. We see little effect on the model's ability to predict the correct answer. Error bars are 95% confidence intervals, model is Gemma-2-9b.

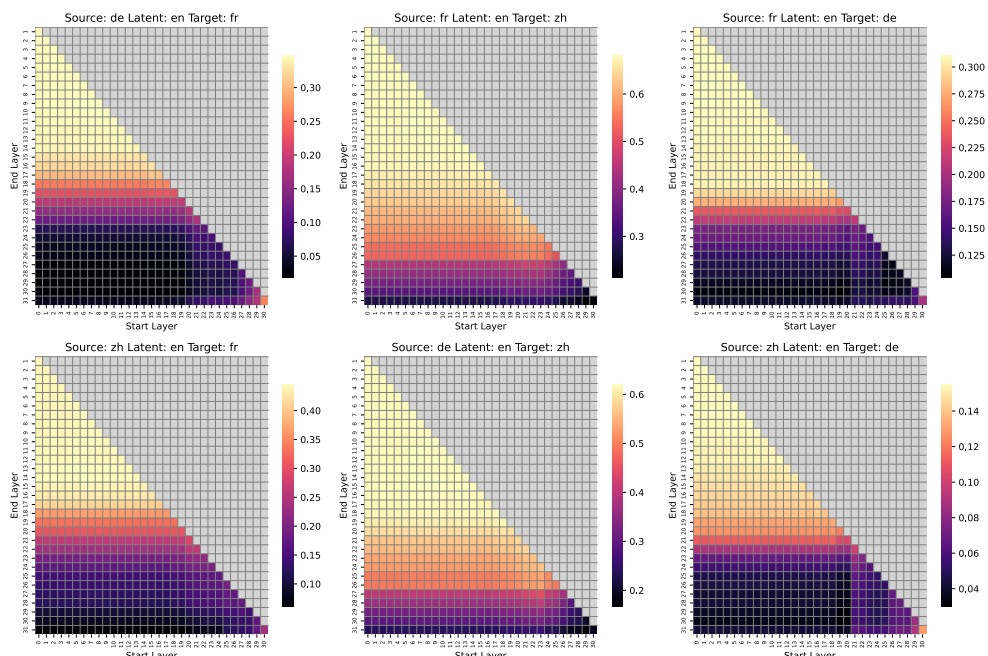

Figure 15: The probability of Llama-2-7b predicting the correct translation, given the rejection intervention on the related English subspace. The intervention was performed across all layers $\{0, 1, \ldots, 31\}$ that lie in the interval (start, end). We sweep over all possible values $0 \leq \text{start} < \text{end} \leq 31$, as well as sweep over the choices for source and target languages. Columns sorted by target language.

FULL EXPERIMENTAL RESULTS FOR STEERING

Here, we focus only on translating from Chinese to French and vice versa, and perform the steering interventions with either German or English as the latent language. Grey regions either indicate an invalid range for the intervention (the start layer is greater than the end layer), or that the residual stream diverged to infinity, which results in a NaN value when fed into the final RMS norm.

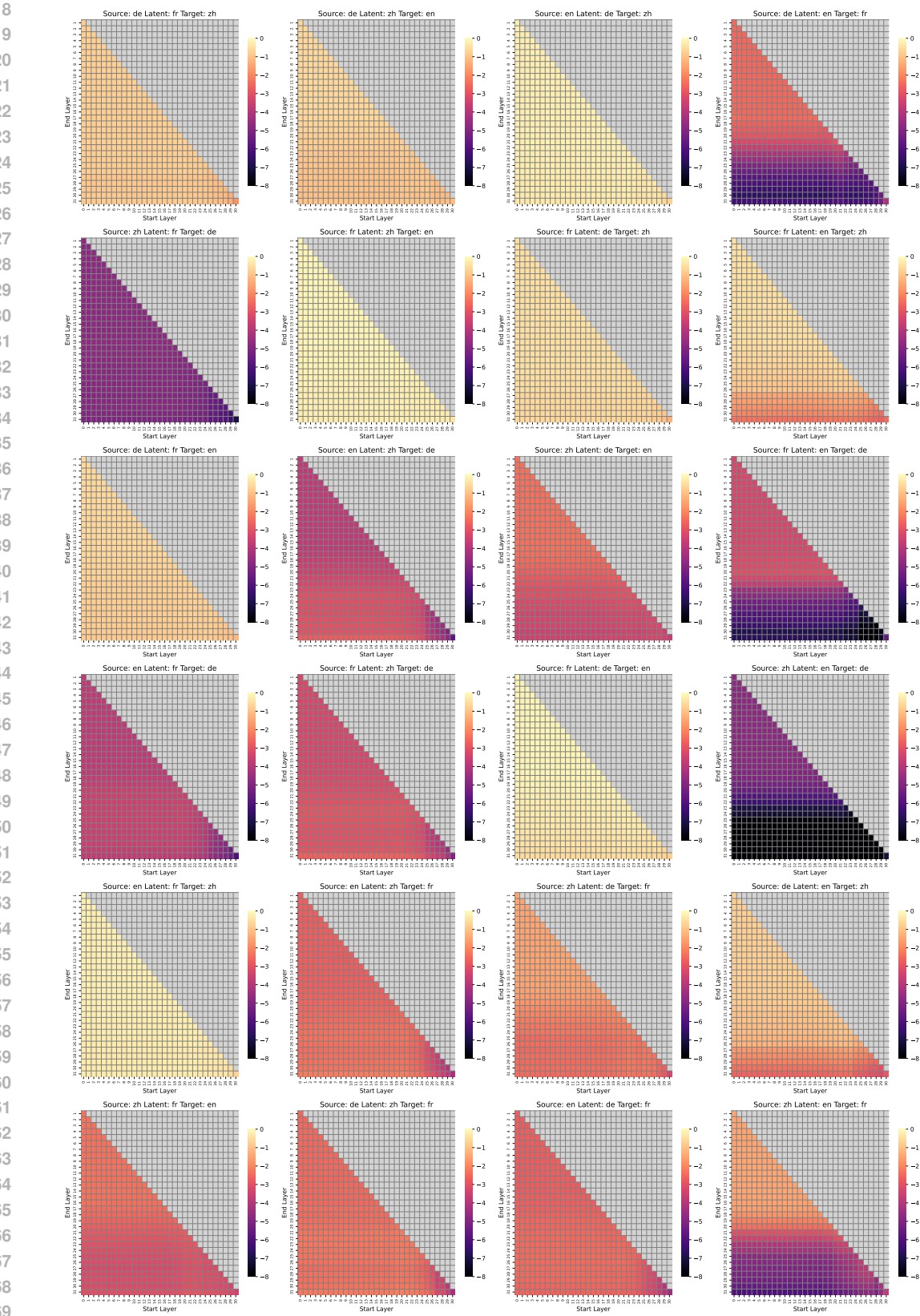

Figure 16: Same plot as Figure 15, but now plotting log-probabilities all over the same range $-8 \leq \log p \leq 0$, and with every choice of latent language. Lower values mean a greater rejection effect. Clearly visible is the greater rejection effect when the latent language is English. Columns sorted by latent language.

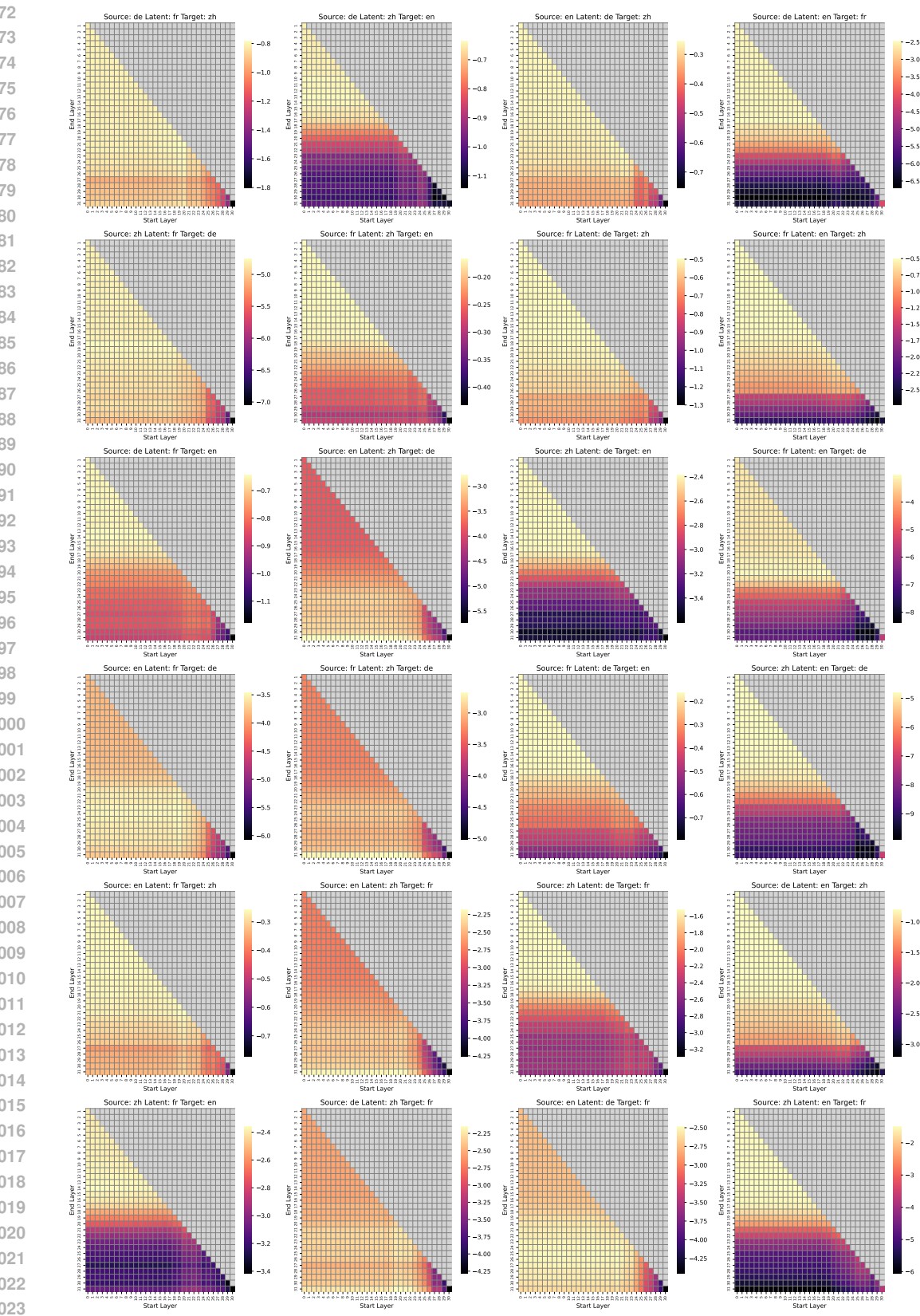

Figure 17: Same plot as Figure 16, but plotting log-probabilities with ranges determined per experiment to make the effect more visible. As before, the range of values is much greater for the English intervention, though interventions in other languages do have a small effect. Columns sorted by latent language.

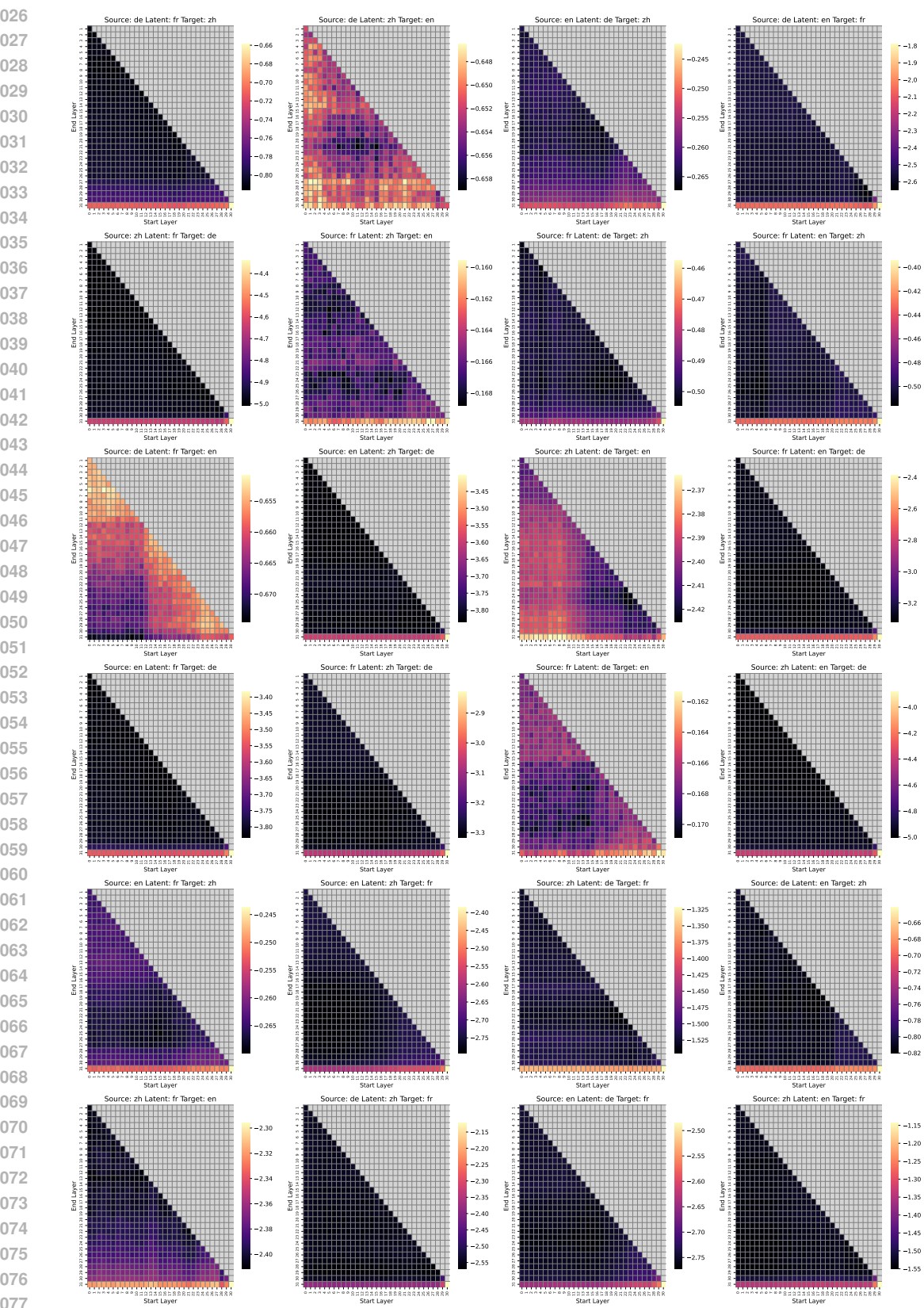

Figure 18: Same plot as Figure 16, but now the intervention is the rejection on the subspace of an unrelated word in the latent language, plotting log-probabilities with ranges determined per experiment to make the effect more visible. This intervention as expected has very little effect on the models prediction.

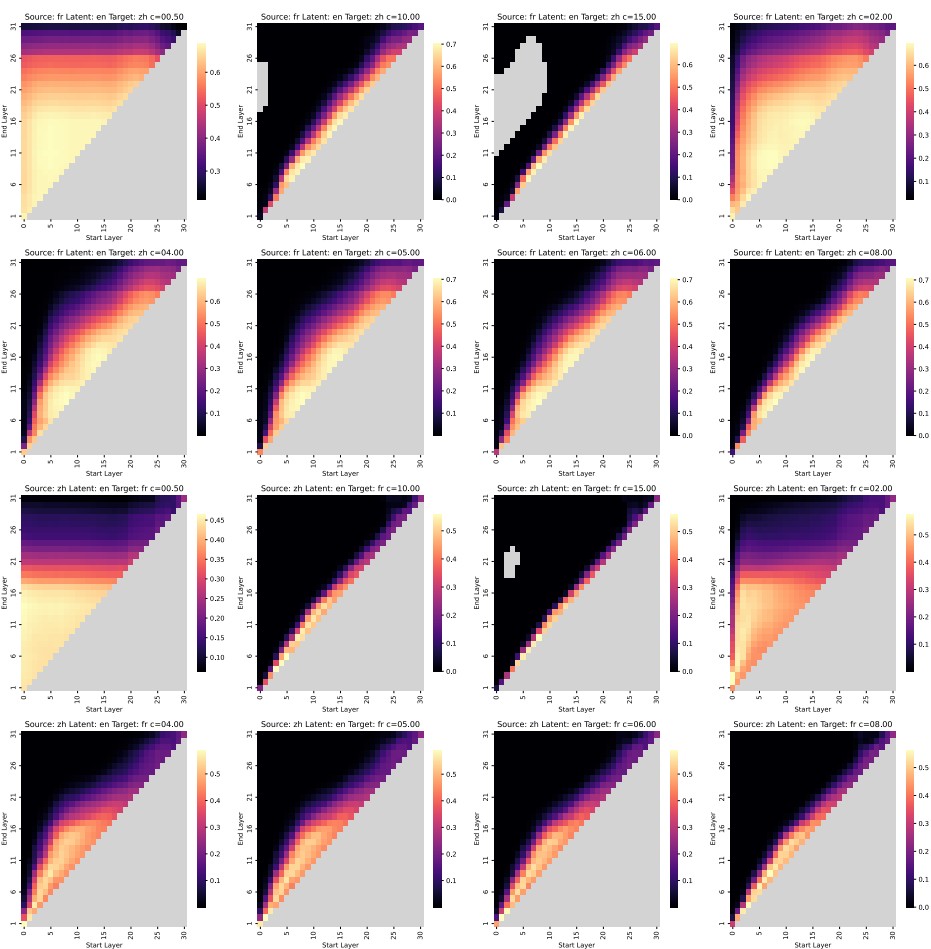

Figure 19: The probability of Llama-2-7b predicting the correct translation, steering towards a different concept in **English**. The intervention was performed across all layers $\{0, 1, \ldots, 31\}$ that lie in the interval (start, end). We sweep over all possible values $0 \leq \text{start} < \text{end} \leq 31$, as well as sweep over the choices for source and target and intervention languages. Lower values indicate a greater steering effect, as we want to steer the model away from predicting the correct translation.

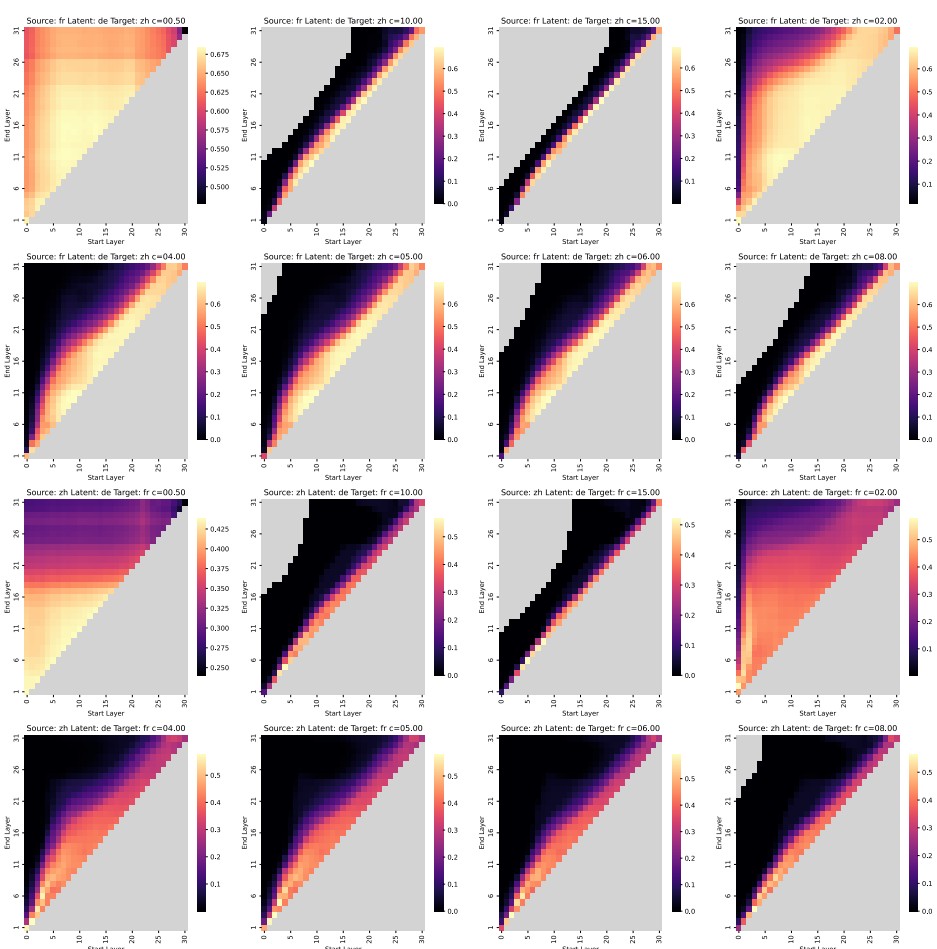

Figure 20: The probability of Llama-2-7b predicting the correct translation, steering towards a different concept in **German**. The intervention was performed across all layers $\{0, 1, \ldots, 31\}$ that lie in the interval (start, end). We sweep over all possible values $0 \leq$ start $<$ end $\leq 31$, as well as sweep over the choices for source and target and intervention languages. Lower values indicate a greater steering effect, as we want to steer the model away from predicting the correct translation.

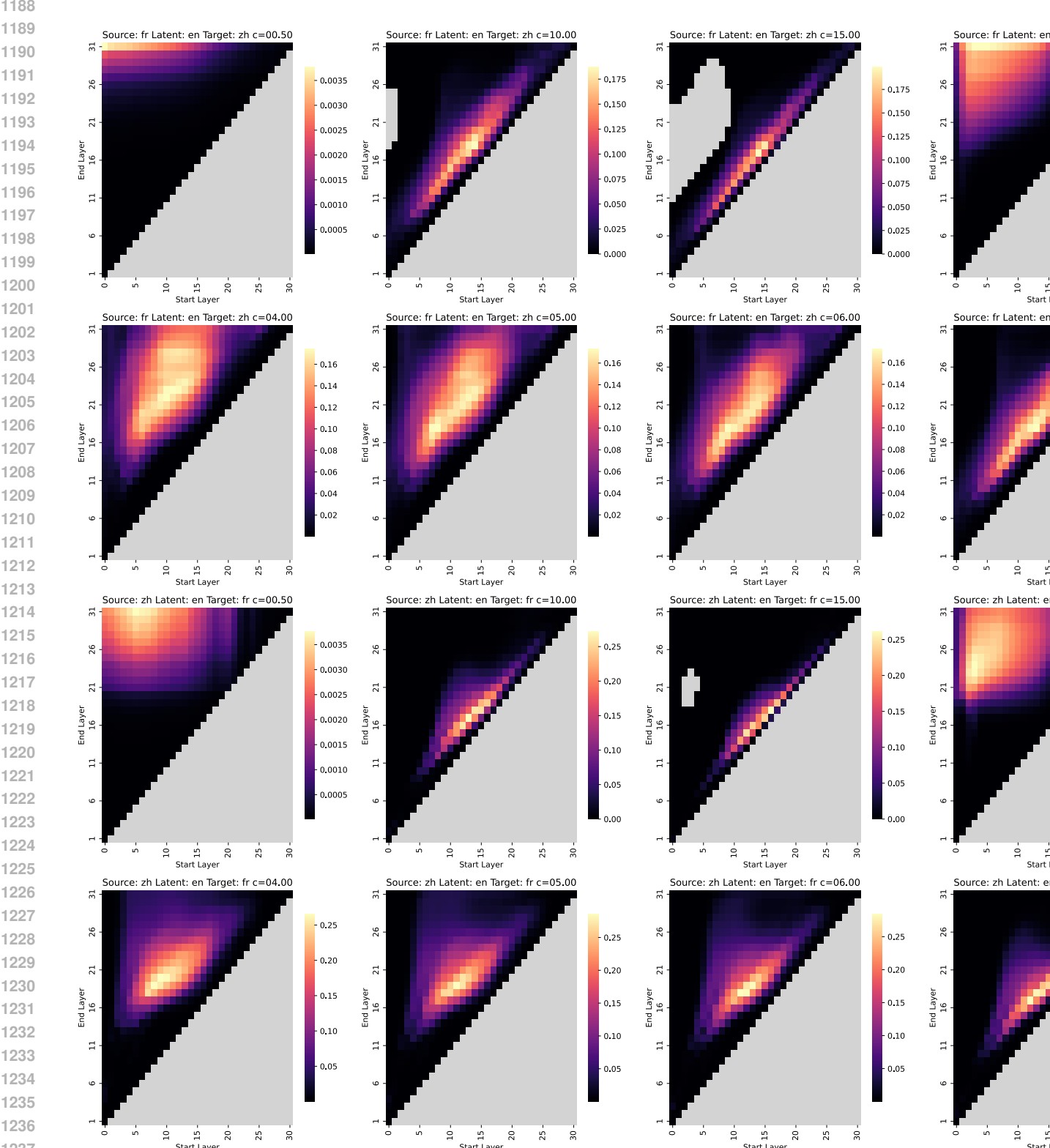

Figure 21: Same plot as Figure 19, but now plotting probabilities of the counterfactual answer we are steering towards, using **English** as the latent language. Higher values mean a greater steering effect.

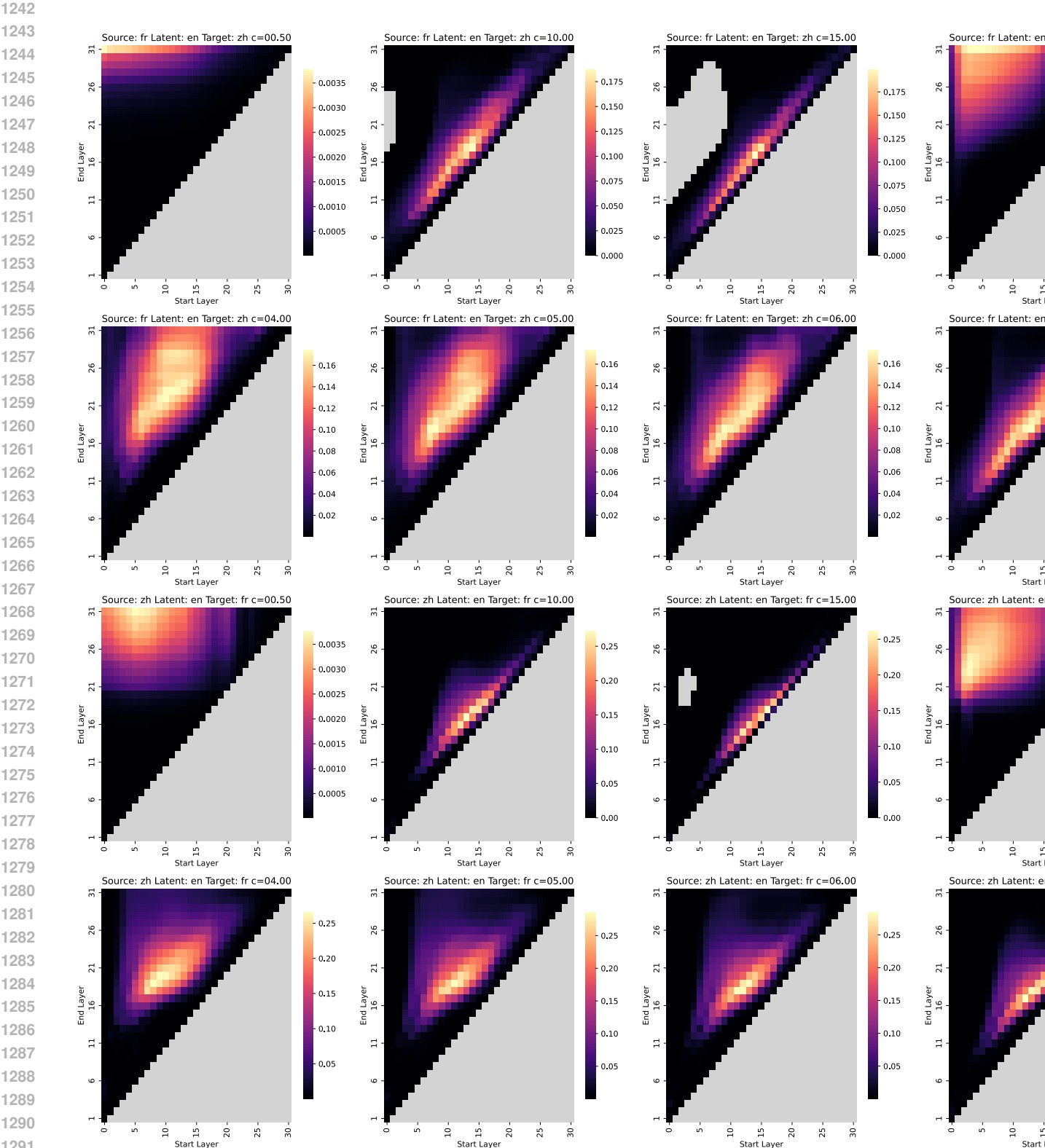

Figure 22: Same plot as Figure 20, but now plotting probabilities of the counterfactual answer we are steering towards, using **German** as the latent language. Higher values mean a greater steering effect.

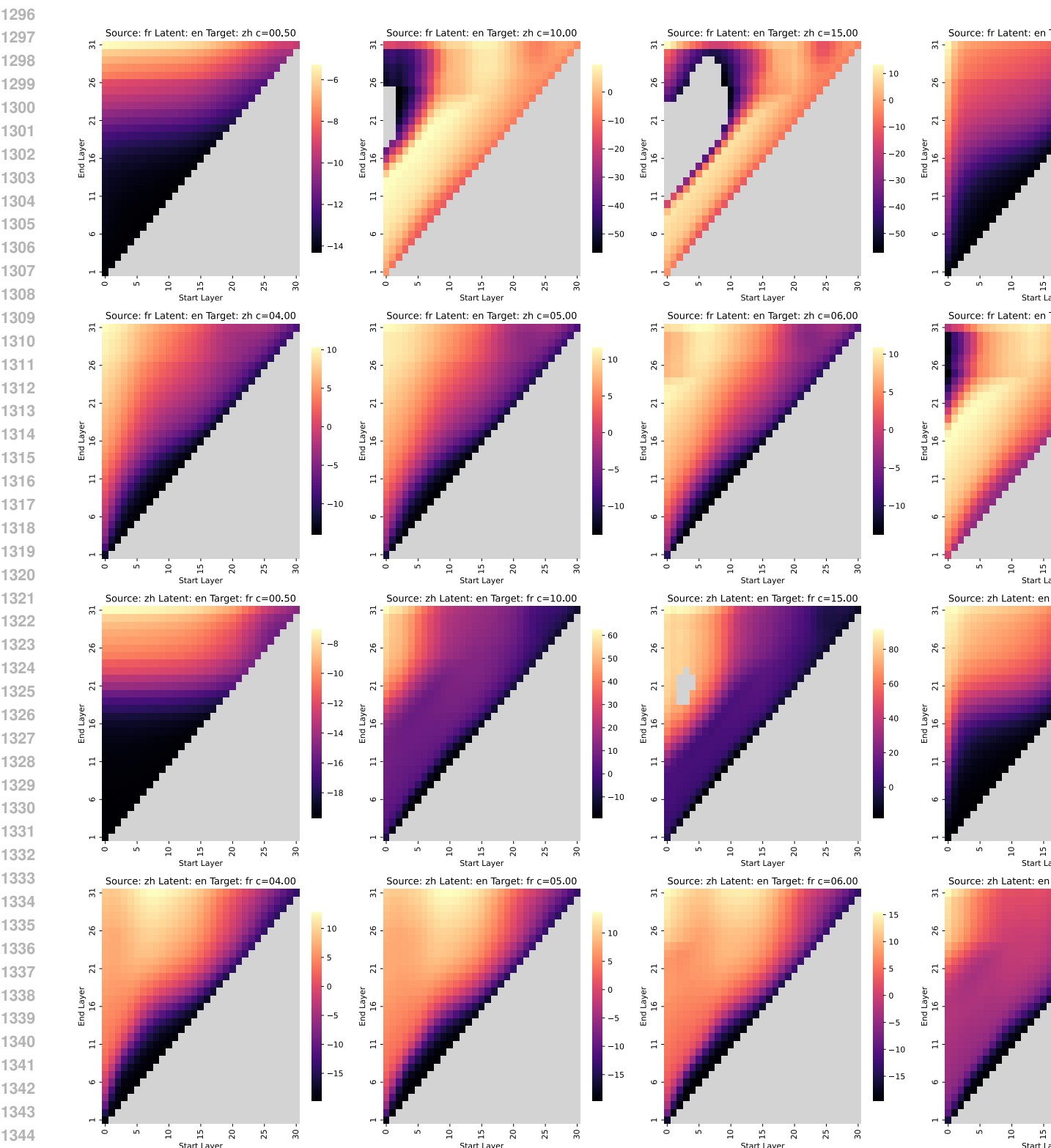

Figure 23: Same plot as Figure 19, but now plotting the log-probabilities of the counterfactual answer we are steering towards, minus the log-probabilities of the correct answer we are steering away from. Intervention is performed with **English** as the latent language. Higher values mean a greater steering effect.

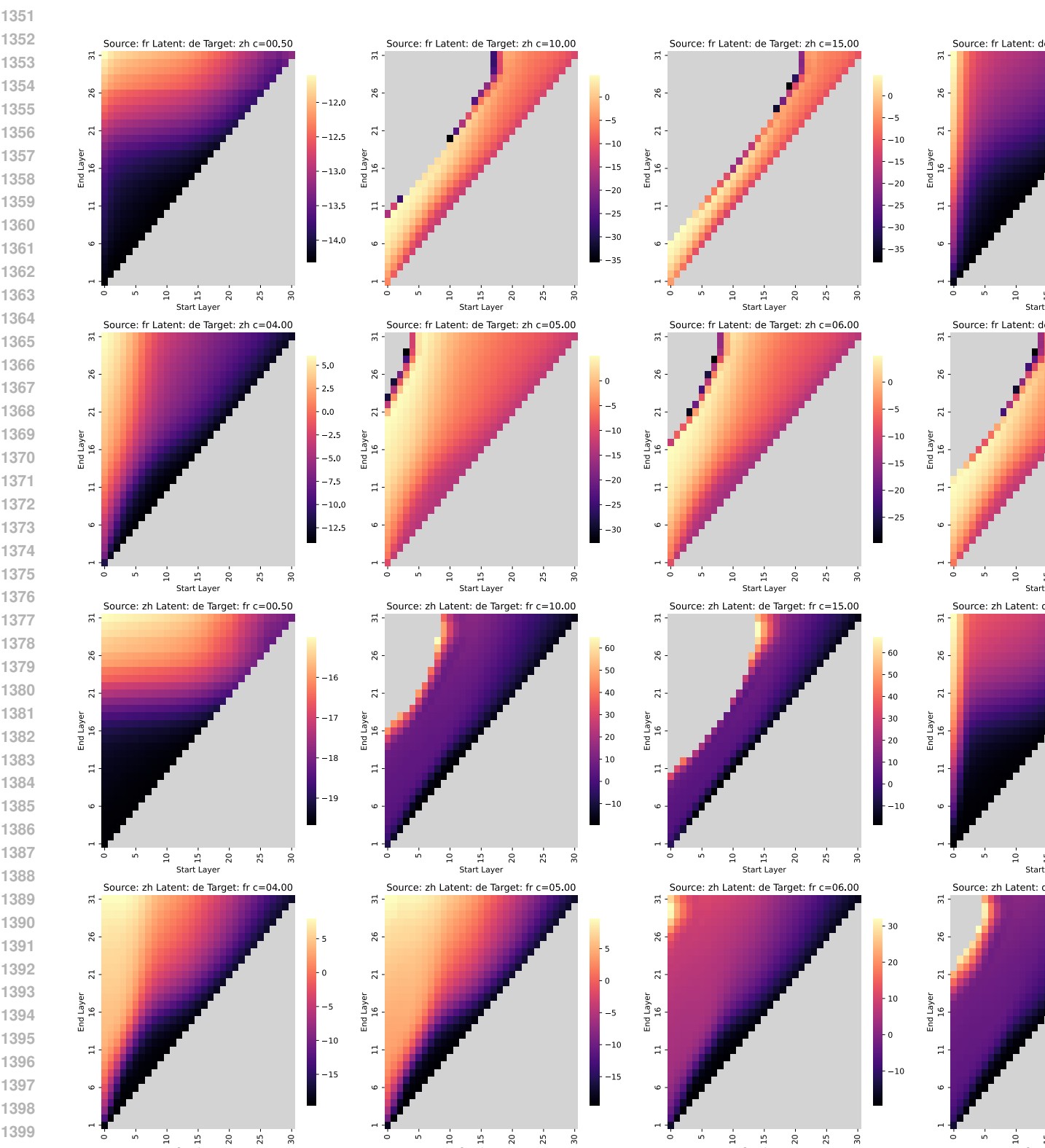

Figure 24: Same plot as Figure 23, but now with **German** as the latent language.

