# OpenReview forum: "Llamas (mostly) think in English: On Causal Interventions in the Latent Language of Transformers"
_ICLR.cc/2025/Conference — Submitted to ICLR 2025_

### Official Review · Reviewer_A1pj · 2024-10-23

**Soundness:** 2
**Presentation:** 3
**Contribution:** 2
**Rating:** 3
**Confidence:** 4

**Summary:**

This paper studies how LLAMA-2 performs computation in a latent language when prompted to translate from one language to another. The paper builds on the surprising findings from Wendler (2024), which showed that residual stream vectors carry information in a English when a model is prompted to translate from one non-English language to another non-English language. The present paper performs causal interventions to test this finding. Given a word to translate, a subspace representing this word is made with unembedding vectors of similar words. Then, the residual stream vector is replaced with its orthogonal projection on its complement, attempting to erase the information related to the subspace. With this setup, the paper shows that erasing English terms from intermediate layers hurts the ability of the model to translate between two other languages. In contrast, erasing other languages as latents does not affect translation quality. And, erasing non-related concepts also does not hurt translations, as a control.

**Strengths:**

1. The question asked by the paper is interesting and important.
2. The question is also reasonably motivated, although the distinction from Wendler's work could be made clearer.
3. The broad methodological approach makes sense, although I have reservations about specifics; see below.
4. The experimental results are convincing, if narrow. In particular, the experiments and results in figures 3+4 are convincing in showing the impact of English as a latent language and non-impact of rejecting unrelated concepts.

**Weaknesses:**

Main weaknesses:

1. The experiments are rather limited. Only one task (translation of simple terms) and only one model.
2. Choice of erasure method is problematic; there may be other ones. See comment below on LEACE.
3. The construction of the subspace with a set of words is somewhat arbitrary, making it hard to draw clear conclusions when the interventions fail to affect performance.
4. Contextualization with prior work is ok but not ideal. Some key references are missing; see below.

See detailed comments in the next section.

**Questions:**

1. The claims in the intro about theory-of-mind capabilities may be exaggerated, see Ullman, 2023, "Large Language Models Fail on Trivial Alterations to Theory-of-Mind Tasks". While the Bubek et al. paper used GPT4, stronger than GPT3.5 used il Ullman's work, the principal issue may remain. So, some modesty in phrasing may be appropriate.

2. Section 2 preliminaries:
- The motivation section is too broad. Mechanistic interpretability is by now quite broad. What specifically is relevant to the present work and what isn't?
- The related work mentioned omits some earlier references. Zhang and Nanda (2024) did not introduce activation patching; rather, they talk about best practices. Probably the first work to apply this technique for interpreting neural LMs is Vig et al., (2020), "Investigating gender bias in language models using causal mediation analysis"", and it's ultimately coming from much earlier work on causal mediation analysis from Pearl and others. "Steering vectors" is a recent popular name to intervention techniques that have been used in the past; the earliest I know is in Bau et al. (2019), "Identifying and Controlling Important Neurons in Neural Machine Translation", see section 6, where they have a similar formulation to that used here, but there might be earlier work.
- The Transformers subsection has too many details that are irrelevant to the work. Since the work only looks at residual stream vectors, the details of the transformers can be omitted or moved to an appendix for completeness.

3. Subspace rejection (section 3.1): this is equivalent to concept erasure, where there's quite a broad body of work, which can be cited. One relevant work is LEACE, by Belrose et al., which provides an optimal linear erasure. Their form is a bit different from the one defined in equation 6, and given their theoretical optimality guarantee, should probably be preferred. It may be that with their formulation the results will be more conclusive.  LEACE provides references to much other related work.

4. Where do the semantically similar words come from? Does it matter how many are chosen for each example? Could the mixed results be explained by the choice of words to use?


5. The heatmaps in figure 5 are hard to read. Using a single-color heatmaps may be better with a baseline of no-intervention indicated somehow, or some other baseline as appropriate for each subplot. Figure 5d would be better as a bicolor colorbar, with 0 as the mid-point. The text in section 3.2 can do a better job explaining how to read figure 5 and interpret it.

6. Formatting: When a citation is not part of a sentence, it should be in parentheses (Smith et al., 2024). In contrast, "Smith et al. (2024) wrote a manuscript about style"".

---

### Official Review · Reviewer_Dfk7 · 2024-10-29

**Soundness:** 2
**Presentation:** 1
**Contribution:** 2
**Rating:** 1
**Confidence:** 4

**Summary:**

This paper seeks to understand the cross-lingual inner workings of Llama 2 by using mechanistic interpretability on text in a number of non-English langauges. In order to do so, they attempt to remove English conceptualization from the activations (hidden states) after each transformer layer in an LLM and evaluate how that affected the model's ability to still perform the same task. They do this removal, or rejection, by creating an English subspace defined as the span and using orthogonal projections to remove the component of the representation that is in this English subspace. When doing so, they find that performance drops significantly. In contrast, when they do the same "rejection" for non-English languages, they find a much less significant drop.

**Strengths:**

The authors come up with a novel mechanistic interpretability scheme to investigate claims from previous papers that recent English-centric LLMs "think in English".
* the methodology is clever and has potential to be effective.
* The paper finds a significant difference when applied to English, versus other languages, presenting a curious phenomena that would be important to be explained.
* The methodology was described well and in detail.

**Weaknesses:**

* The paper, generally speaking, is poorly organized. It doesn't adhere to typical conventions of academic papers in this field, and as a result is difficult to follow.
* A lot of valuable information is placed in the appendix and barely referenced in the main body of the text.
* The "motivation" doesn't adequately argue for the importance of the work and it s never really motivated.
* In many specific instances, insufficient background is provided before using technical terms or even arbitrary terms chosen by the authors
* A significant lack of references to both provide context for the work amongst the current state of the field (e.g. Related Work section) as well as severely lacks justification for many of the claims and design choices. There are 0 citations in the last 5 pages out of 8 despite there being a tremendous amount of content.
* Their findings are not at all replicated when moving to a different model (Gemma-2), undermining the significance of their findings on Llama 2. This lack of transferability implies the possibility that the significant results (like those presented in Figure 3) are caused by some other unexplained phenomena unrelated to Llama "thinking in English".
* The results are presented really poorly. Some visualizations (notably Figure 5) confused me more than supported claims they made. While graphs had CIs, very few actual numerical results were provided (say, in table form), undermining both reproducibility and detail.
* The writing conveys the notion that the authors did not set up proper experiments and rather found results & then came up with alternatives to see if the original results were significant. A potential takeaway is that the authors might have been "searching for a conclusion" and then found results to justify them, as opposed to properly implementing the scientific method.

My intuition is that their results on Llama 2 are an artefact of something else, and that I do not see enough justification to back up their arguments & conclusions. The method itself, of using orthogonal projections, I have a hard time believing would lead to reasonable conclusions about how LLMs "think". I think the span of specific token representations would not be a good "English subspace".

I believe the weaknesses of this paper to be too substantial to overcome its contributions, and recommend a rejection.

**Questions:**

* I would advise adding more to the abstract to give a fuller idea of what will be in the paper
* I would suggest reorganizing Section 1.1 on Wendler et al, 2024 to Section 2 and creating a more developed "Related Works" section
* I would suggest moving motivation to intro

* Why go into so much detail about the architecture of transformers? So much of it does not seem relevant to your paper and a significant amount can be assumed to be understood by any LLM researcher.
* Line 174: what argument can be made for transformers ?
* Line 194: provide explanation & justification for such a hypothesis, it does not make to sense as to why it would be so simple all while being a core part of your methodology.
* Line 250: tying the effectiveness of the rejection to the part of the model that Wendler et al. 2024 states is where the model "thinks in English" is very much not explained or grounded.
* Figure 3 & 4: keep order of graphs in same order of bars (DE, EN, ES, FR, RU, ZH)
* Explanation of Figure 5 needed. I spent a lot of time attempting to dissect it and it still does not make sense to me as to why Figure 5a is so clean (and reversed ?). In addition, Figure 5 is never thoroughly discussed in the text.
Section 3.4 (Future Work) uses too casual of language and does not adequately back up it's analysis of Gemma, nor does it provide sufficient motivation for the second paragraph. This section reads more as a "Limitations" section
What's the point of Figure 2 ?

---

### Official Review · Reviewer_B44e · 2024-11-09

**Soundness:** 3
**Presentation:** 1
**Contribution:** 2
**Rating:** 5
**Confidence:** 4

**Summary:**

This paper builds on recent work that argues that, when prompted in a non-English language, LLMs likely perform an implicit translation to English in their intermediate representations; task-relevant computation then occurs on top of these intermediate representations, finally translating back to the relevant target language. The authors try to tighten this claim by performing interventions and demonstrating causality (at least partially) of the intermediate English representations to the ability to solve the task at hand. The results for the presented task are fairly convincing, but I have a few apprehensions due to (i) use of a singular task, (ii) use of perhaps too strong an intervention protocol, and (iii) general organization issues in the paper.

**Strengths:**

Prior work's claim on use of English as the "concept space" by LLMs for performing their reasoning was a fairly intuitive claim, but making the claim rigorous requires establishing causality. This paper's target is thus practically really important, since an English bias may render models anglocentric. The experiments are relatively straightforward and easy to follow, with fairly well written explanations (the general presentation of the paper needs quite a bit of work though).

**Weaknesses:**

Broadly, I have three apprehensions about the paper and hence currently lean towards rejecting it. I believe these issues can be addressed relatively easily, and likely during the rebuttals phase itself. If they are addressed, I am happy to raise my score.

- **Use of a singular task.** From what I can gather, there is only one task studied in the paper: translation of a concept / word from a source language to a target language. This task is indeed sufficient as a proof of concept, but given the breadth of the claim this paper is trying to verify (i.e., "LLMs 'think' in English"), I would argue more scientific rigor is warranted. To that end, simply repeating the tasks studied in the prior work this paper is building upon (Wendler et al., 2024) should suffice. Trying other tasks may also open the door to more creativity, e.g., by allowing an intervention derived from one task to be used on another one.

- **The intervention protocol may be too strong.** The studied task is almost entirely about producing specific tokens. In my experience using logit lens in such tasks, and often even in open-ended generation, the hidden representations at intermediate layers have a really large cosine similarity with the unembedding layer of the model. Especially for the single token generation task used in this paper, it seems likely this behavior is at play because Figure 5 suggests translation accuracy is quite high after Layer 1 itself. Correspondingly, the ablation used in this paper---i.e., removing information from a subspace defined using the ground truth answer and other closely related tokens---may in fact be removing the representation's project along the token's embedding. This is then bound to reduce model's performance, and could also explain why the counterfactual experiments work well. I think a simple experiment to help address this concern would be evaluating cosine similarity of the intermediate representations and the ground truth answer token's (or related tokens) embedding. The similarity can be evaluated with English tokens, but also their translations. If the similarity is high with respect to English token embeddings, even though the source language and target language were not English, then that would give some evidence that regardless of the protocol being overpowered, there is unlikely to be any confounding.

- **General organization and presentation.** The paper looks and reads a bit rushed. For example, Figure 1 is quite blurry; there is basically no related work section (including in the appendix, and despite the main paper being only 8 pages); poorly organized figures (Figure 5 is references before Figures 2--4, but occurs two pages later; this is probably because the figure is large, but I really don't see why the four assembled panels in that figure must be presented simultaneously, or why they should be so big); citation commands are incorrectly used (\citet and \citep should be used depending on whether the reference goes in parenthesis or not); appendix is quite sparse and crucial seem missing details (e.g., I don't know what Figures 16--24 are about, since the text never refers to them, and it's also unclear which models the figures correspond to; also, Figures 21--24 go out of margins). Overall, quite a bit of presentation work is needed. If the authors can manage to do this during rebuttals, I'd be willing to re-read the paper; however, I do not think the paper can be published in its current presentation state.

**Questions:**

For Figure 5, what do results look like without any intervention? The reject intervention seems to work well relatively later in the model, but if it had any effect in earlier layers too, it's hard to know that because it's unclear what one should compare results to. An alternative to providing the baseline (i.e., no-intervention accuracy) would be plotting change in accuracy, instead of absolute performance (as is currently the case in plots).

---

### Meta-Review · Area_Chair_jTSK · 2024-12-04

**Metareview:**

This paper attempts to test whether all reasoning in the model requires English conceptualization. They identify an English subspace, and they remove it from the model basis, demonstrating that this removal significantly damages performance. Based on these results, they claim that the models rely on English conceptual thinking.

Reviewers appreciate the objective of using a causal intervention to test existing claims about model reliance on English.

Overall, the actual experiments were not considered sufficiently robust or rigorous. In particular, there are alternative reasons why the English subspace might involve more semantics than subspaces for lower frequency languages, especially with the particular intervention protocol used.They also had some presentational issues

**Additional Comments On Reviewer Discussion:**

Likely because of the universally negative reviews, the authors did not engage in a rebuttal.

---

### Decision · Program_Chairs · 2025-01-22

Reject